# Amino Acid Digestibility of Different Formulations of Torula Yeast in an In Vitro Porcine Gastrointestinal Digestion Model and Their Protective Effects on Barrier Function and Inflammation in a Caco-2/THP1Co-Culture Model

**DOI:** 10.3390/ani13182812

**Published:** 2023-09-05

**Authors:** Lynn Verstrepen, Marta Calatayud-Arroyo, Cindy Duysburgh, Jelle De Medts, Ricardo D. Ekmay, Massimo Marzorati

**Affiliations:** 1ProDigest BV, Technologiepark 82, 9052 Zwijnaarde, Belgium; lynn.verstrepen@prodigest.eu (L.V.); cindy.duysburgh@prodigest.eu (C.D.); jelle.demedts@prodigest.eu (J.D.M.); 2Center for Microbial Ecology and Technology (CMET), Faculty of Bioscience Engineering, Ghent University, Coupure Links 653, 9000 Ghent, Belgium; 3Institute of Agrochemistry and Food Technology (IATA-CSIC), Spanish National Research Council, 46980 Valencia, Spain; 4Arbiom Inc., Durham, NC 27703, USA; rekmay@arbiom.com

**Keywords:** torula yeast, *Cyberlindnera jadinii*, wood hydrolysate, gastrointestinal digestion, pig, amino acid, inflammation, cytokines, intestinal barrier, in vitro

## Abstract

**Simple Summary:**

Pork meat consumption accounts for about 30% of worldwide meat consumption and its production process is known to have an environmental impact. Hence, targeting animal nutrition is an effective way to reduce the environmental impact of pig farming, and the use of alternative protein sources are appealing to supply the required building blocks to maintain pig health. One of these alternative protein sources is torula yeast. Using a simulation of the stomach and upper part of the gastrointestinal tract of pigs, this study showed that different manufacturing processes of torula yeast had an impact on the availability of the required building blocks for pig growth. In addition, after digestion, the ability to improve gut health in a cell culture model was affected by differences in cultivation and manufacturing of the protein source. Overall, this study provided novel insights into the mechanisms behind the health improvement of pigs fed with torula yeast and the importance of manufacturing processes for alternative protein sources.

**Abstract:**

Single-cell protein from torula yeast (*Cyberlindnera jadinii*) grown on lignocellulosic biomass has been proven to be an excellent alternative protein source for animal feed. This study aimed to evaluate the amino acid (AA) digestibility by estimating intestinal absorption from three yeast-based ingredients, produced by cultivating *C. jadinii* on hydrolysate, using either mixed woody species (drum- (WDI) or spray-dried (WSI)) or corn dextrose (drum-dried (DDI)) as the carbon source. Further, the protective effect of intestinal digests on activated THP1-Blue™-induced epithelial damage and cytokine profile was evaluated. Total protein content from these three ingredients ranged from 34 to 45%, while the AA dialysis showed an estimated bioaccessibility between 41 and 58%, indicating good digestibility of all test products. A protective effect against epithelial-induced damage was observed for two of the three tested products. Torula yeast cultivated on wood and drum-dried (WDI) and torula yeast cultivated on wood and spray-dried (WSI) significantly increased transepithelial electrical resistance (TEER) values (111–147%, *p* < 0.05), recovering the epithelial barrier from the inflammation-induced damage in a dose-dependent manner. Further, WSI digests significantly reduced IL8 (250.8 ± 28.1 ng/mL), IL6 (237.9 ± 1.8 pg/mL) and TNF (2797.9 ± 216.3 pg/mL) compared to the blank control (IL8 = 485.7 ± 74.4 ng/mL, IL6 = 478.7 ± 58.9 pg/mL; TNF = 4273.5 ± 20.9 pg/mL) (*p* < 0.05). These results align with previous in vivo studies, supporting torula yeast-based ingredients as a high-quality protein source for pigs, protecting the intestinal barrier from inflammatory damage, and reducing the pro-inflammatory response. We provided novel insights into the mechanisms behind the health improvement of pigs fed on torula yeast-based ingredients, with potential applications for designing nutritional interventions to recover intestinal homeostasis during critical production periods, such as weaning.

## 1. Introduction

The main goal of swine production is to efficiently convert feedstuffs into pork meat with the highest safety, nutritional, and organoleptic properties for human consumption [1], with minimal environmental impact, and at the lowest cost. Despite a high-performance level of animal production, advances in sustainability are still required [2]. Animal feeding, including the crop cultivation phase, manufacturing process, and transportation, is the main contributor to the environmental impact associated with the pig production system [3]. Pork meat consumption accounts for the 30% of worldwide meat consumption [4], and progress in feed development, precision feeding, and manure management is required to reduce the climate impact per kg of meat produced [5]. Therefore, animal nutrition is an effective target to reduce the environmental impact of pig farming, as it can affect land use for crop production, emissions, and manure compositions [6].

Protein is one of the major nutrients required in pig production systems, mainly for the supply of the required amino acids (AA) which are essential building blocks for protein synthesis [7]. The efficiency of dietary protein use is determined by the digestibility of the protein, the intestinal absorption, and the potential interactions with other feed constituents or the gut microbiota [7]. Thus, estimation of bioaccessibility (i.e., the fraction of AA that become accessible for absorption) [8] and bioavailability (i.e., the proportion of AA that reach systemic circulation and can be incorporated into body protein synthesis) [9] are key parameters to determine diet quality and to adjust the balance between AA supply and AA requirements [10]. AA have multiple functions beyond protein synthesis, including immune regulation, improvement of intestinal epithelial barrier functions, acting as signaling molecules that regulate mRNA translation, or being precursors of neurotransmitters (e.g., γ-aminobutyrate, dopamine, and serotonin), small peptides (e.g., glutathione and carnosine), low-molecular-weight nitrogenous metabolites (e.g., NO, polyamines, and catecholamines), carbon monoxide (CO), and H2S, which encompass diverse and important physiological functions [1,11]. Thus, a precise AA supply is required for optimal performance at different steps of the swine cycle production, from the breeding to finishing stages, while avoiding excess in AA supply contributes to reducing the environmental impact, minimizing the nitrogen excretion as urea [7]. 

Weaning is considered one of the most critical periods in pig production, associated with changes in small intestinal morphology and function, digestion and absorption capacity disruption, intestinal barrier impairment, and potential infections and growth retardation [12]. Gut health significantly affects overall health status and nutrient utilization. The intestinal epithelium is involved in the final digestion and absorption of nutrients and the secretion of protective mucins, and it forms a barrier against harmful antigens and pathogens [1]. Therefore, nutritional intervention to recover the intestinal barrier integrity during the weaning period and beyond can be a strategy to reduce weight loss and improve the health status of piglets. As AA are critical molecules in maintaining intestinal homeostasis [1], investigation of the effects of complex intestinal digests based on dietary AA on the epithelial barrier and intestinal inflammation is of interest. 

As an alternative to conventional protein sources, torula yeast (*Cyberlindnera jadinii*) has a high value because it can be cultivated on lignocellulosic waste biomass, does not rely on agricultural land use, has independent geographical cultivation, and has proven to be an excellent source of highly digestible AA [13,14,15,16]. It has been effectively incorporated in diet formulations for cats [17], poultry [18], pigs [15,16,19], and multiple aquaculture species [20,21]. Torula yeast can assimilate pentoses, including xylose and arabinose, and other carbon sources, such as organic acids, alcohols, propionaldehyde, and acetaldehyde. The nutritional composition of torula yeast-derived biomass depends on the composition of the fermentation medium used in yeast production and the downstream processes, which can ultimately affect the nutrient availability and functionality of the yeast [22]. Active dry yeast is the most common form of yeast supply in feed, but it can also be provided inactivated or fractionated, though at increased production costs. When administered live, yeast can act as a probiotic, improving gut health and growth performance [23]. Moreover, yeast biomass is a good source of macroelements such as calcium, phosphorus, and zinc, micronutrients such as selenium, and chromium or B-complex vitamins [24]. The yeast cell wall consists of different polysaccharides, including mannan oligosaccharides and β-glucans [21,25], both molecules affecting immune homeostasis and intestinal health [26,27,28]. Previous studies have shown that *C. jadinii* can improve animal intestinal health, maintain growth performance compared to conventional protein sources, and modulate microbiota composition, oxidative status, and intestinal homeostasis [29,30,31,32]. However, the role of production conditions, including carbon source and downstream processing on torula yeast AA profile and intestinal health modulation, is still underexplored. 

Thus, this study estimated, in vitro, the AA bioaccessibility and AA absorption of three iterations of torula yeast-derived ingredients produced on different carbon sources (dextrose vs. wood) and subject to drum-drying or spray-drying processes using a porcine gastrointestinal digestion model. The effect of these intestinal digests on the epithelial barrier and immune modulation of the gut was also assessed in an in vitro human Caco-2/THP1-Blue™ co-culture model, providing novel mechanistic insights on dietary modulation of intestinal pig health. 

## 2. Materials and Methods 

All reagents were provided by Merck KGaA, Darmstadt, Germany, unless otherwise stated. 

### 2.1. Product Description

This research tested three torula yeast products. To produce the ingredients, two lots of torula yeast were produced by cultivating *Cyberlindnera jadinii* on hydrolysate (LC hydrolysate 1, LC1) derived from mixed woody species (maple, poplar, ash, and beech) as the carbon source. A third lot of torula yeast was produced by cultivating *C. jadinii* on corn dextrose as the carbon source. Both cultivations were provided with mineral salts to meet nutritional needs and incubations were performed following identical procedures. Briefly, cultivations were started as shake flasks. After achieving the target optical density after approximately 18 h, a 40 L bioreactor was inoculated with 10% of the volume from the shake flasks. After approximately 18 h, 10% of the broth volume was transferred into a 1000 L bioreactor. The fermentation continued as a fed-batch until a 24 h-effective fermentation time was achieved. Yeast broth was concentrated, washed with 3× distilled water to resuspension, reconcentrated, and pasteurized. Half of the broth produced from wood-derived hydrolysates was drum-dried (WDI) and half was spray-dried (WSI). The lot of torula yeast based on dextrose was drum-dried (DDI). Further, a second wood hydrolysate (LC2) from the same woody species was used as control in the cell culture assays (described in Section 2.6). Products were provided by SylPro, Arbiom Inc., Durham, NC, USA. Proximate composition of yeast products is described in Table 1.

### 2.2. Testing Conditions

Two independent experiments were carried out during this research. 

In the first experiment, a screening of the three ingredients was performed. 

Torula yeast samples were digested at a dose of 3.2 g/100 mL, conducted as detailed in Section 2.3. Samples were collected before the digestion procedure (raw product) and at the end of the small intestinal digestion and dialysis procedure for total amino acid quantification and amino acid profiling (estimated AA absorption). Further, samples were collected at the end of stomach digestion (Stend) and at the start (70 min; SIstart), mid (140 min; SImid), and end (210 min; SIend) of the small intestine digestion and were used for further cell culture tests after a centrifugation step (9000× *g*, 5 min) and 60% (*v*/*v*) dilution in cell culture media to evaluate the effect of partially digested/absorbed fractions along the gastrointestinal tract. The comparative effect with the pure products at equivalent doses was performed.

In a second experiment, WDI was selected for assessment of a dose-response, using samples from SIend, following an identical digestion and dialysis procedure. Different doses were assayed (3.2 g, 1.6 g, 0.8 g, and 0.4 g/100 mL) to determine the minimal effective in vitro dose on the selected endpoint. DDI was used as a negative control, based on previous results. All assays were performed in triplicate. 

### 2.3. Upper Gastrointestinal Simulation 

The adult porcine in vitro gastrointestinal digestion model comprised a three-compartmental system, including oral, gastric, and small intestinal digestion, each in technical triplicates. The mastication step was simulated by mincing the food in an electric mincer (Bosh ErgoMixx, 750 W, potency 4, 30 s). Initially, tested products were weighted in individual reactors (3.2 g) and filled with simulated salivary fluid in a proportion 1:2 (*w*/*v*), containing KCl 15.1 mM, KH_2_PO_4_ 3.7 mM, NaHCO_3_ 13.6 mM, MgCl_2_(H_2_O)_6_ 0.15 mM, (NH_4_)_2_CO_3_ 0.06 mM, HCl 1.1 mM, CaCl_2_(H_2_O)_2_ 0.025 mM, and 50 U/mL α-amylase. Oral phase digestion was maintained for 2 min, 37 °C, and shaking (300 rpm) (MaxQ 4000 Benchtop Orbital Shaker, Thermo Fisher Scientific, Merelbeke, Belgium). Subsequently, gastric digestion was conducted by adding 54.6 mL protein- and carbohydrate-depleted gastric fluid containing 6.9 mM KCl and 47.2 mM NaCl, 320 µL lecithin solution (0.17 mM), and 1450 µL 2% pepsin solution. Then, a sigmoidal pH reduction was implemented by adding HCl 0.5 M, starting at pH 4.5 and reaching pH 1.8 at the end of the gastric digestion (120 min). Continuous pH control during the gastric phase was performed by a Senseline pH meter F410 (ProSense, Oosterhout, The Netherlands) and an automatic pump dosage of HCl (0.5 M) or NaOH (0.5 M). 

After the stomach incubation, the gastric digestion volume was measured and adjusted to 70 mL, and small intestinal digestion was carried out. During the small intestinal incubation, 30 mL of simulated intestinal fluid (SIF, pH 7) was added, with a final concentration of 0.525 mM CaCl_2_(H_2_O)_2_, α-amylase 0.07 mg/mL (35 U/mL), lipase 0.09 mg/mL (>1800 U/mL), chymotrypsin 0.2 mg/mL (>8 U/mL), trypsin 0.04 mg/mL (40–80 U/mL). Simultaneously, pancreatic juice components were added to the simulated gastric fluid to a final concentration of 44.9 mM NaHCO_3_ and 0.598 mg/mL bile salts (Oxgall).

Digestive fluids were homogenously mixed, and a duodenal phase without dialysis was conducted for 30 min, pH 6.1–6.4, 37 °C, with shaking at 300 rpm (Cole-Parmer™ Stuart™ Orbital Shaker, VWR International, LLC, Leuven, Belgium). After the duodenal phase, the jejunal and ileal digestion was started by quantitatively transferring (~100 mL) duodenal fluid into a dialysis bag of regenerated cellulose with a molecular weight cut-off of 3.5 kDa, 28.6 mm diameter, 60 cm long (Carl Roth). Dialysis bags were clamped and placed on flasks containing dialysis fluids. The digestion was conducted for 210 min, with replacement of dialysis solution every 70 min, and pH was maintained in the range of 6.4–6.7 at 37 °C, with shaking at 300 rpm. The intestinal lumen:dialysis solution ratio was 1:2 (*v*/*v*). Samples were obtained at different incubation times, including at the end of the gastric digestion and small intestinal digestion at 70, 140, and 210 min, including the remaining non-absorbed fraction (inner compartment of the dialysis tube) for amino acid quantification and dialysate fluid for cell culture experiments. 

Amino acids absorption was estimated by a quantitative differential approach, calculated as: Absorbed AA (mg)=AA in the raw sample−AA in the non−absorbed fraction

### 2.4. Hydrolysis Procedures

Products were subjected to three different procedures, including acid hydrolysis, performic acid oxidation followed by acid hydrolysis, and alkaline hydrolysis, to cover all the AA spectra.

For the different procedures, 200 µL of the sample were mixed with the corresponding hydrolysis solution: 6 M HCl for acid hydrolysis, performic acid solution (90% *v*/*v* formic acid BioUltra, 1.0 M in H_2_O and 10% hydrogen peroxide 30% (*v*/*v*) for performic acid oxidation, and 8 M NaOH for alkaline hydrolysis. 

Subsequently, acid and alkaline hydrolysis were incubated for 24 h at 105 °C and dried under an N_2_ stream at 50 °C. Performic acid oxidation was incubated for 1 h at 50 °C, dried under an N_2_ stream at 50 °C, and 2 mL 6 M HCl was added, followed by incubation for 24 h at 105 °C and drying under an N_2_ stream at 50 °C. 

Then, samples were dissolved in water (1:1 *v*/*v*) containing 250 µM internal standard (aminobutyric acid (GABA)) and derivatized using the ACCQ-FLUOR reagent kit (Waters, Antwerp, Belgium), following manufacturer instructions. 

Amino acids were measured using the HPLC method and included aspartic acid, glutamic acid, serine, histidine, glycine, threonine, alanine, proline, cysteine, tyrosine, valine, methionine, lysine, isoleucine, leucine, phenylalanine, and tryptophan.

The acid hydrolysis allowed detection of all AA except cysteine, methionine, and tryptophan. The performic acid oxidation was applied to quantify cysteine and methionine, and alkaline hydrolysis was used to quantify tryptophan. 

As a remark, the method to determine the total amino acid content (requiring acid hydrolysis of the protein) transforms glutamine and asparagine into their corresponding acids, i.e., aspartic acid and glutamic acid. Therefore, aspartic acid also includes asparagine, while the total amount of glutamic acid also includes glutamine.

### 2.5. High-Performance Liquid Chromatography with Diode-Array Detection

High-performance liquid chromatography with diode-array detection (HPLC-DAD; Hitachi Chromaster HPLC-DAD, VWR, Leuven, Belgium) was conducted with the three hydrolysates. Briefly, chromatographic separations were achieved on a pre-heated (37 °C; 5310 Column oven) column Luna Omega, Polar C18, 150 × 4.6 mm, 3 um, 100 Å (Phenomenex Inc., Utrecht, The Netherlands), with the mobile phase consisting of acetonitrile HPLC Plus, ≥99.9% (solvent A), water containing 140 mM sodium acetate and 17 mM trimethylamine (pH = 5.05) (solvent B), and ultrapure water (solvent C) in different proportions (5160 Pump). Mobile phase gradients are described in Appendix A. The flow rate was set at 1 mL/min, and samples (5 µL) were injected with an autosampler. Detection was conducted at 250 nm excitation/395 nm emission (5430 DAD).

### 2.6. Cell Culture

Caco-2 cells (HTB-37; American Type Culture Collection) were maintained in Dulbecco’s modified eagle medium (DMEM), containing glucose (4.5 g/L) and glutamine (0.6 g/L), and supplemented with HEPES (10 mM) and 20% (*v*/*v*) heat-inactivated (HI) fetal bovine serum (FBS) (Gibco, Thermo Fisher Scientific, Merelbeke, Belgium). THP1-Blue™ NF-κB reporter cells were obtained from InvivoGen (Toulouse, France) and maintained in Roswell Park Memorial Institute (RPMI)1640 medium, containing glucose (2 g/L) and glutamine (0.3 g/L), supplemented with HEPES (10 mM), sodium pyruvate (1 mM), and 10% (*v*/*v*) HI-FBS. Cells were incubated at 37 °C in a humidified atmosphere of air/CO_2_ (95:5, *v*/*v*). The Caco-2/THP1-Blue™ co-culture was performed as previously described [33]. Briefly, Caco-2 monolayers were cultured for 14 days on 24-well semi-permeable supports, until a functional cell monolayer was obtained. Before starting the co-culture, TEER values were measured (0 h). Then, 48 h before the start of the co-culture, THP1-Blue™ cells were seeded in 24-well plates and stimulated for 48 h with phorbol 12-myristate 13-acetate (PMA; 100 ng/mL). After PMA removal, Caco-2 cells were placed on top of PMA-differentiated THP1-Blue™ cells. Then, centrifuged (5 min, 9000× *g*) upper gastrointestinal suspensions, diluted to 60% in complete cell culture medium, were added to the apical compartment. A control condition (cell culture media) was included in each run. Further, torula yeast pure products (DDI, WDI, WSI; 19.2 mg/mL) and two wood hydrolysate controls [LC hydrolysate 1, LC1; LC hydrolysate 2, LC2; 0.3 mg/mL], dissolved in water and diluted to 60% in complete cell culture media, were tested. After 24 h of incubation, TEER was measured as a marker of epithelial barrier integrity, and 24 h TEER values were normalized to their own 0 h value and presented as a percentage of the initial value. Further, the basolateral medium was removed and THP1-Blue™ was stimulated with ultrapure lipopolysaccharide (LPS) from *Escherichia coli* K12 (InvivoGen, Toulouse, France; 500 ng/mL) or left untreated. After 6 h of the pro-inflammatory challenge, basolateral medium was collected for cytokine quantification (human IL-6, IL-8, IL-10, TNF-α, CXCL10) (Luminex Technology, Thermo Fisher Scientific, Merelbeke, Belgium) and NF-κB measurements (QUANTI-Blue reagent, InvivoGen, Toulouse, France). 

### 2.7. Statistical Methods

Different AA compositions between products was analyzed using a two-way ANOVA with Tukey’s multiple comparisons tests.

For the first cell culture experiment, statistically significant differences between the treatments and their respective controls were calculated using an ordinary one-way ANOVA with Tukey’s multiple comparisons tests. For the second part, statistically significant differences between the treatments and the control were calculated using an ordinary one-way ANOVA with Dunnett’s multiple comparisons tests. Significant differences are marked with asterisks: (*), (**), (***), and (****) represent *p* < 0.05, *p* < 0.01, *p* < 0.001, and *p* < 0.0001, respectively.

Statistical analysis and graphs were performed with GraphPad Prism software 8.3.0 for Windows (GraphPad Software, San Diego, CA, USA).

## 3. Results

### 3.1. Total Amino Acid Content and Amino Acid Profile of Torula Yeast Products

The protein content and the AA profile of test products are displayed in Table 2. The protein content was determined at 42.7 g/100 g for DDI, 44.7 g/100 g for WDI, and 33.6 g/100 g for WSI, with highest content thus observed for WDI reaching significance compared to the other test products (*p* < 0.0001). For three products, the most abundant amino acids, in absolute numbers and relative to the total protein, were glutamic acid (Glu, 14.3–17.5%), methionine (Met, 14.1–18.5%), and aspartic acid (Asp 9.4–10%). WDI showed a higher relative abundance of alanine (Ala, 8.1%) compared to DDI and WSI (6.5–6.9%), reaching statistical significance (*p* < 0.0001). The most abundant amino acid was Met for WSI (*p* < 0.0001 compared to both other products), and Glu for DDI (*p* < 0.0001 compared to WSI) and WDI (*p* < 0.0001 compared to WSI). DDI had the highest relative levels of tryptophan (Trp) (7.6%), reaching significance compared to WSI (6.4%; *p* < 0.0001) and WDI (4.3%; *p* < 0.0001). Histidine (His) and lysine (Lys) were the least abundant amino acids, with only minor differences observed between the different test products. Indeed, for His, no significant differences were observed between the products (*p* > 0.9999), showing relative abundances of 1.5%. For Lys, relative abundances ranged between 1.7 and 1.8%, with DDI showing a significantly reduced Lys proportion as compared to WDI (*p* = 0.0102) and WSI (*p* = 0.0102).

During upper gastro-intestinal incubation, the estimated absorbed fraction of protein was the highest for DDI (57.7%), followed by WDI (52.3%; *p* = 0.1144 compared to DDI) and WSI (41.4%; *p* < 0.0001 and *p* = 0.0023 compared to DDI and WDI, respectively) (Figure 1), indicating a good digestibility of all test products. DDI and WDI showed similar total AA content in the dialyzed fluid (787.9 mg/reactor and 749.3 mg/reactor, respectively) (Table 3), though still reaching statistical significance (*p* < 0.0001), whereas WSI showed a significantly lower AA content (445.2 mg/reactor) (*p* < 0.0001) compared to both other products (Table 3). 

Considering specific amino acids, Met, Glu, and Trp were detected at the highest levels in the dialyzed fluids from the three products. Met was especially high in WSI (34.9%), reaching significance compared to WDI (23.6%; *p* < 0.0001) and DDI (18.9%; *p* < 0.0001) (Table 3) when considering the percentage to total protein content in the digestive fluid. 

Contrarily, Glu and Trp showed the highest percentages in DDI (17.2% and 13.2%, respectively), reaching significance compared to the other products, expect for Glu, when compared to WDI (for Glu: *p* = 0.5591 and *p* = 0.0095 compared to WDI and WSI, respectively; for Trp: *p* = 0.0343 and *p* = 0.0006 compared to WDI and WSI, respectively) (Table 3). Remarkably, Thr showed a low content in the WSI diet (0.8%) compared to DDI and WDI (3.1–3.2%), though not reaching statistical significance (*p* = 0.4643 and *p* = 0.4340 compared to DDI and WDI, respectively) (Table 3).

When calculating the percentage of individual AA dialysability, compared to each AA level in the raw diet, Trp was the only AA reaching 100% dialysability in DDI and WDI diets, with only 37.4% of Trp dialysate in the WSI diet (*p* < 0.0001 compared to both other products). Thr also showed marked differences between diets, with values of 39.0–34.0% (DDI and WDI, *p* = 0.3045 between these diets) compared to 7.4% in WSI (*p* < 0.0001 compared to both other products).

### 3.2. Pure Products and Proximal Gastrointestinal Digests Are More Effective in Protecting the Epithelial Barrier Integrity

Samples at the end of the stomach and different time points of small intestinal digestion were tested on the simulated intestinal epithelium to evaluate the effect of digestion processes on product modulation of the epithelial barrier. As controls, pure products and two wood hydrolysate controls were included (LC1, LC2). DDI did not significantly affect TEER values compared to the respective control conditions (Figure 2). Remarkably, WDI and WSI pure products and St_end_ and SI_start_ digests significantly increased TEER values (101–147%, *p* < 0.05) compared to corresponding controls. Mid and end small intestinal digests from the three products did not show significant differences compared to the control condition. 

### 3.3. Effect of Ingredients Digests on Intestinal Inflammatory Response 

Gastrointestinal digests from the dialysed fraction at the end of the small intestinal digestion were tested for their effects on LPS-induced pro- and anti-inflammatory cytokine secretion in the Caco-2/THP1-Blue™ co-culture model of the gut. 

DDI and WDI were the products affecting fewer cytokines, with the only significant reduction observed for IL6 (DDI, 303.6 ± 12.9 pg/mL; WDI, 249.2 ± 11.7 pg/mL) compared to the control condition (478.7 ± 58.9 pg/mL) (*p* < 0.05) (Figure 3). In contrast, WSI reduced most of the cytokines, including IL8 (250.8 ± 28.1 ng/mL), IL6 (237.9 ± 1.8 pg/mL), and TNF-α (2797.9 ± 216.3 pg/mL) compared to the control (IL8 = 485.7 ± 74.4 ng/mL; TNF-α = 4273.5 ± 20.9 pg/mL) (*p* < 0.05). CXCL10 was not significantly affected by any of the treatments; however, WDI induced a slight reduction (*p* > 0.05).

NF-κβ activity was reduced by the three products when compared to LPS or control condition, but did not reach significant values (*p* > 0.05) (Figure 3).

Confirming previous results, TEER was significantly increased by WDI and WSI compared to the control condition, whereas DDI did not affect TEER (Figure 3). 

### 3.4. WDI Confers a Dose-Dependent Protection of the Epithelial Barrier

The dose-response effect of the WDI diet on inflammation-induced epithelial damage is shown in Figure 4. WDI digests significantly increased TEER at a dose of 3.2 g (159.3 ± 6.8%) and 1.6 g (143.6 ± 5.4%) (*p* > 0.05). When supplemented at 0.4 g, WDI digests induced a significant decrease in TEER (101.9 ± 2.1%) compared to the control (117.6 ± 2.2%). The effect of DDI digests on the epithelial barrier was similar to the control condition (115.7 ± 0.3%), as observed in previous experiments. 

## 4. Discussion

This in vitro study demonstrated that torula yeast protein sources provide a balanced AA supply upon entering the small intestine. Yeast intestinal digests had a differential protective effect against inflammation-induced epithelial damage and immunomodulation, depending on carbon source and suggesting the applicability of torula yeast to promote pig health through nutritional interventions.

The protein source used in this research originated from wood biowaste or corn dextrose through transformation by yeast (*Cyberlindnera jadinii*), comparable to previous studies applied in aquaculture and pet feeding [15,16,17,19]. Overall, DDI and WDI had a closer composition profile than WSI, with higher assessed values of estimated protein digestibility obtained through measurement of AA absorption during the in vitro digestion procedure (around ~52–58%). In vivo, total protein digestibility would probably be even higher [19], as digested small peptides could still be further broken down by brush border enzymes (which were not included in the current in vitro study). The spray-drying process to produce WSI is a differential parameter compared to DDI and WDI, which were drum-dried during the production process. Compared to spray-drying, the lower surface and larger particle size during drum-drying have been described as influential factors in protecting the stability of food bioactives [34], likely also affecting AA in this study. 

Met and Cys also showed high dialysability (70–78%). Met and Cys are essential and conditionally essential AA, respectively, with significant roles in swine health. Cys and Met are both sulfur AA, involved in maintaining intestinal homeostasis. Met is the second or third limiting amino acid in typical swine diets [35,36] and its deficiency negatively affects pig growth performance. An adequate intake of dietary methionine is required to maintain intestinal mucosal integrity, morphological development, and intestinal antioxidant capacity, while the activated form of Met, S-Adenosylmethionine, is involved in multiple metabolic processes [37]. Cys is involved in the immune control of the intestinal epithelium, participates in the biosynthesis of glutathione, a potent anti-oxidant molecule, and plays a crucial role in attenuating local inflammation and restoring gut homeostasis [37,38]. 

Glu and Trp were, after Met, the AA with the highest percentage of dialysability. Remarkably, Trp was the only AA dialyzed at 100% (compared to the Trp level in the product) in DDI and WDI, with significantly lower values in WSI (~37%). The differences in the production process of WSI (spray-dried vs. drum-dried) could influence Trp solubility, a hypothesis that would require further studies. Trp has been included in the classification of “functional amino acids”, referring to those AA that improve the efficiency of utilization of dietary proteins in pigs [38], and deficits in Trp have been linked to immune alterations and oxidative stress [39]. Recently, increased Trp concentration was proven to increase indole-3-lactic acid and 3-indoleacetic acid, potent immunomodulatory bacterial metabolites, and regulates bile and nitrogen metabolism in two pig gut lactobacilli species [40]. 

Glu is the most abundant free AA in the body, with multiple functions on pig health. Glu is an important energy source for intestinal epithelial cells, affecting intestinal development, structure and function, and growth performance of weaning piglets. During weaning, glutamine concentrations are significantly reduced in the intestinal fluid (70%), tissue (−38%), and plasma (−30%) [41]. Glu levels in our intestinal dialysate would range from 16 to 42 g/kg, again with higher levels for DDI. Similarly, other AA provided with torula yeast-derived products, such as Arg, Gly, Thr, Cys, and Pro are gatekeepers of intestinal health, attenuating intestinal damage, maintaining barrier integrity, restoring immune homeostasis, and balancing inflammatory responses in the gut [1,37,42]. 

To mechanistically address the effect of porcine small intestinal dialysates on the intestinal barrier and inflammation, we used a human in vitro model of activated macrophages-induced epithelial damage and inflammation as proof-of-concept. Remarkably, the DDI did not have a protective effect on inflammation-induced intestinal epithelial damage compared to WDI and WSI. DDI was based on torula yeast growth on dextrose, compared to the wood-based material that was a carbon source in WDI and WSI. As the AA profile is similar between DDI and WDI, these findings suggest that other wood-derived bioactives, different than AA, could play a role in preserving intestinal barrier integrity. However, the wood hydrolysates themselves elicited no protective effect. Further, the yeast product is separated from the growth medium and washed prior to drying; this suggests that the wood hydrolysates may induce the production of bioactive molecules, rather than a carryover effect from the hydrolysates themselves. DDI had lower fat levels than WDI and WSI, another factor to further study as a cause of the observed differences. In that sense, protein content in WSI was lower than in the other ingredients, suggesting that protein level or dialysable AA content, though not deficient, is not the triggering factor in protecting the epithelial barrier from inflammatory damage. 

The protective effect on inflammation-induced epithelial barrier disruption was higher in non-digested samples and proximal small intestinal digests, suggesting that the duodenal and jejunal sections can be more influenced than ileal and colonic sections after ingredient intake. Potentially, digestive enzymes and longer incubation times can affect the bioactive molecules responsible for the protective effect on the epithelial barrier, inactivating them in the simulated porcine small intestinal digests at the end-time point. These results suggest a protective effect of porcine intestinal digests at anatomical locations, with more susceptibility to environmental clues and a higher immunomodulatory capacity [43]. The jejunum (reviewed in [44]) is the larger section of the small intestine (88–91%) [45] dedicated to nutrient absorption. Gut-associated lymphoid tissues (GALT) are present in the small intestine, with higher proportion of T cells in the jejunal Peyer patches (PP) and a higher T/B cells ratio compared to ileal PP [46]. Pathogenic infections occur more prevalently in the small intestinal segments, where pathogenic bacteria can attach, colonize, and cause infection, and where the pressure for niche competition with gut bacterial commensals is lower than in the distal gut regions. This is the case of Enterotoxigenic *Escherichia coli* (ETEC), one of the leading causes of postweaning diarrhea in newly weaned pigs [47]. Therefore, the protective effect of torula yeast-based products on small intestinal integrity can reinforce immune system and swine health by releasing bioactive molecules during the digestion process, which have a modulatory effect on the epithelial barrier at proximal sites. 

Additionally, we observed a dose-dependent effect of products on epithelial barrier protection, suggesting a minimal feed intake (1.6 g in vitro, corresponding to an estimated 400 g in vivo) to observe the beneficial effects. A previous study showed that increasing torula yeast levels in the diet positively affected nutrient utilization and inflammatory response [15], indicating a dose-response effect of torula yeast, supported by our in vitro results. 

Differential effect of digests from torula-based products was also observed on the cytokine profile. DDI and WDI dialysate had a lower impact on cytokine profile than WSI, only reducing IL6 levels, whereas WSI had a potent anti-inflammatory potential, reducing TNF-α, IL8, and IL6. These results are consistent with in vivo assays, reporting a linear reduction of TNF-α as the concentration of torula yeast increased in the diets of weanling pigs [15]. Besides AA supply, torula yeast also contains immunomodulatory and prebiotic molecules, including polysaccharides, mannoproteins, chitin, and β-glucans from the cell wall [19]. β-glucans are potent stimulators of the immune system, affecting macrophages, T helper cells, neutrophils, and natural killer cells, leading to pathogen resistance via humoral and cellular immunity [28]. In a pig intestinal model, yeast cell walls containing β-glucans reduced CD22 receptors to reduce inflammation, with different responses observed depending on the yeast processing technique [48]. As WSI showed the highest impact on immune modulation, the effect of spray-drying processes could partially affect torula yeast properties. Previous research has shown that *Saccharomyces cerevisiae*-derived glucans had a higher immunomodulatory activity after spray-drying than lyophilization or solvent exchange, suggesting that spray-dried preparations are preferred for the use of (1→3)-β-D-glucan as immunomodulator/adjuvants in the form of aqueous suspension [49]. 

Supporting our observations in vitro, Cruz et al. recently reported that a feeding strategy for weaned piglets, in which 40% of crude protein in feed was replaced with torula yeast grown on lignocellulosic biomass, maintained growth and improved digestive function, including intestinal villi height and dry fecal matter [16]. 

In conclusion, our research aligns with previous in vivo studies, supporting torula yeast-based ingredients as high-quality protein feeds for pigs, protecting the intestinal barrier from inflammation-induced damage and reducing the local proinflammatory response. We observed that different carbon sources used for yeast growth, dietary fat content, and industrial production processes (i.e., drum-drying vs. spray-drying) could influence AA digestibility, as well as the effect of intestinal digests on gut health. Concretely, wood-derived torula yeast protected the inflamed epithelium and induced a higher anti-inflammatory response than dextran-derived extracts. Whether this effect is due to wood-derived hydrolysates or non-wood-derived conditions (i.e., suboptimal growth conditions) would require further studies. 

## Figures and Tables

**Figure 1 animals-13-02812-f001:**
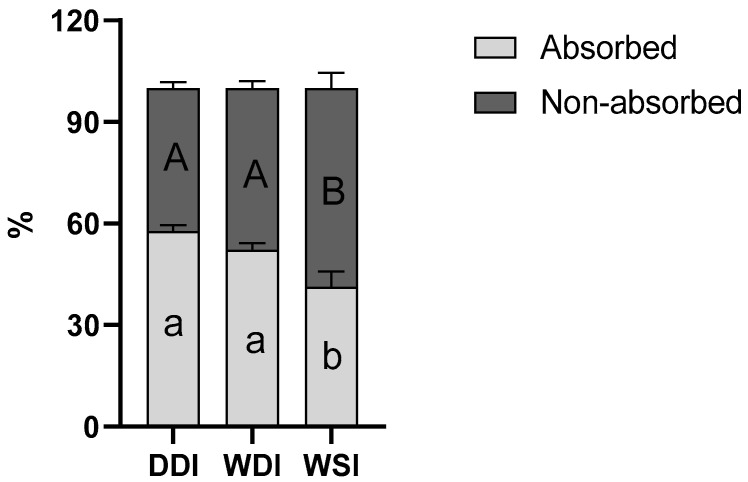
Total amino acid content at the end of the small intestinal incubation. Bars represent the total AA content in the simulated digestive fluid from the inner (non-absorbed) and outer (absorbed) compartment of the dialysis tubular membrane. Measured amino acid levels during the blank incubation were subtracted from measurement during digestion of the different test products. Results are expressed as proportional values (%) (geometric mean ± standard error of the mean; *n* = 3). Statistical analysis was performed to compare the different test products using a two-way ANOVA with Tukey’s multiple comparisons post hoc test. Statistical differences between the test products are indicated with different letters (small letters for absorbed fraction and capital letters for non-absorbed fraction) (*p* < 0.05).

**Figure 2 animals-13-02812-f002:**
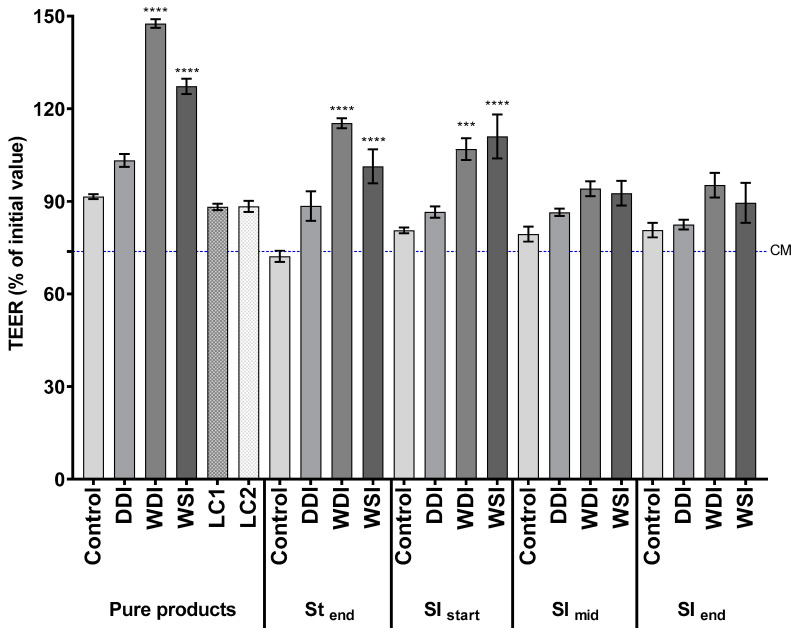
Effect of ingredients and intestinal digests on transepithelial electrical resistance (TEER) of the Caco-2/THP1-Blue™ co-cultures. Bars represent the TEER values measured 24 h after treatment normalized to its corresponding 0 h value (%; mean ± SEM; *n* = 3). The blue dotted line corresponds to the complete medium (CM) control. Statistically significant differences compared to the control condition are marked with asterisks (*** *p* < 0.001 and **** *p* < 0.0001). St = stomach; SI = small intestine; DDI = dextrose-derived drum-dried yeast ingredient; WDI = wood-derived drum-dried yeast ingredient; WSI = wood-derived spray-dried yeast ingredient; LC1 = LC hydrolysate 1; LC2 = LC hydrolysate 2.

**Figure 3 animals-13-02812-f003:**
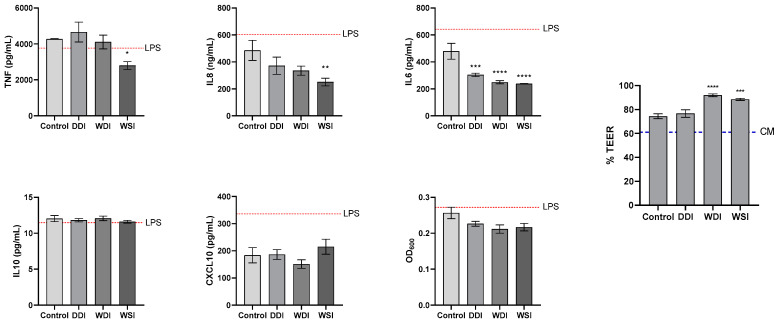
Effect of upper gastrointestinal tract (GIT) samples on epithelial barrier and inflammatory response. TEER, cytokine, and NF-κβ activity levels were measured 6 h after LPS treatment on the basolateral side of the Caco-2/THP1-Blue™ co-cultures after pre-treatment of the apical side for 24 h with the upper GIT samples. The red dotted line corresponds to the experimental control LPS +. The blue dotted line corresponds to the complete medium (CM) control. Data are plotted as mean ± SEM. Statistically significant differences between the treatments and the control condition are marked with asterisks (*), *p* < 0.05 (*), *p* < 0.01 (**), *p* < 0.001 (***), and *p* < 0.0001 (****). DDI = dextrose-derived drum-dried yeast ingredient; WDI = wood-derived drum-dried yeast ingredient; WSI = wood-derived spray-dried yeast ingredient.

**Figure 4 animals-13-02812-f004:**
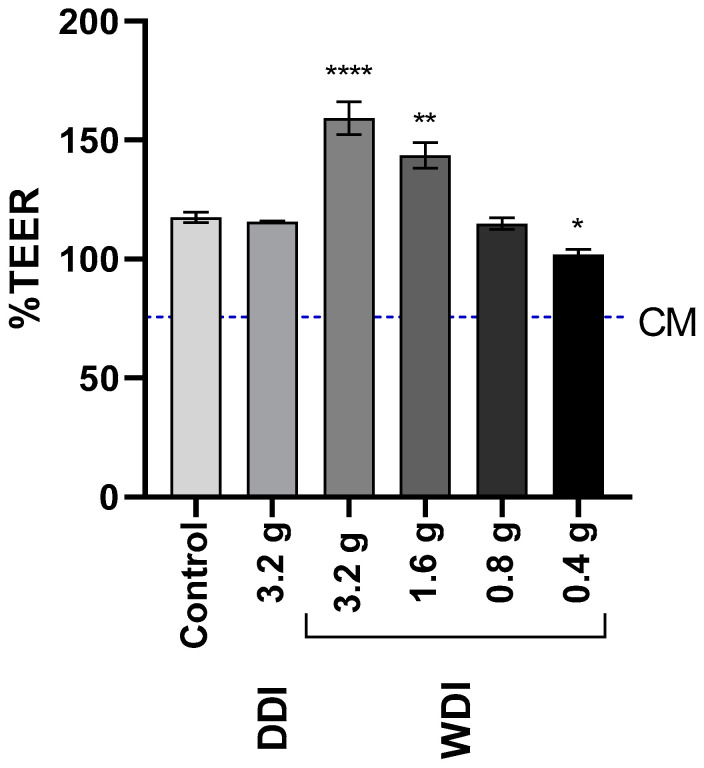
Dose-response effect of upper GIT samples on transepithelial electrical resistance (TEER) of the Caco-2/THP1-Blue™ co-cultures. Bars represent the TEER values measured 24 h after treatment, normalized to its corresponding 0 h value (%; mean ± SEM; *n* = 3). The blue dotted line corresponds to the complete medium (CM) control. Statistically significant differences compared to the control condition are marked with asterisks (* *p* < 0.05, ** *p* < 0.01, and **** *p* < 0.0001). DDI = dextrose-derived drum-dried yeast ingredient; WDI = wood-derived drum-dried yeast ingredient; WSI = wood-derived spray-dried yeast ingredient.

**Table 1 animals-13-02812-t001:** Proximate composition of yeast products used in this study. Detailed compositional analysis of torula yeast used as protein source has been previously reported [19].

	g/100 g			
	Moisture	Crude Protein	Crude Fat	Ash
DDI	5.3	54.8	1.0	10
WDI	5.0	51.0	5.4	9.8
WSI	5.0	46.1	5.2	9.8

**Table 2 animals-13-02812-t002:** Total amino acid (AA) content (mg/incubation, corresponding to mg/3.2 g product) and relative amino acid proportion (%) of different products (drum-dried torula yeast broth produced on dextrose (DDI), drum-dried torula yeast broth produced on wood-derived hydrolysates (WDI), and spray-dried torula yeast broth produced on wood-derived hydrolysates (WSI)). Percentage of individual AA in products are calculated against total AA content. Results are shown as averages ± stdev (*n* = 3). Statistical analysis was performed to compare the different test products using a two-way ANOVA with Tukey’s multiple comparisons post hoc test. Statistical differences between DDI and WDI, and DDI and WSI, are indicated in bold, while significant differences between WDI and WSI are underlined (*p* < 0.05).

	mg AA/Incubation (3.2 g)			% AA		
Amino Acid	DDI	WDI	WSI	DDI	WDI	WSI
Alanine (Ala)	94.08 ± 0.29	**116.51** ± 0.07	**69.54** ± 0.07	6.9 ± 0.02	**8.1** ± 0.01	**6.5** ± 0.01
Arginine (Arg)	58.85 ± 0.10	**61.02** ± 0.13	**38.26** ± 0.08	4.3 ± 0.01	4.3 ± 0.01	**3.6** ± 0.01
Aspartic acid (Asp)	128.70 ± 0.13	**136.32** ± 0.06	**107.35** ± 0.07	9.4 ± 0.02	**9.5** ± 0.01	**10.0** ± 0.01
Cysteine (Cys)	71.50 ± 0.25	**50.82** ± 0.18	**38.52** ± 0.17	5.2 ± 0.02	**3.5** ± 0.01	**3.6** ± 0.02
Glutamic acid (Glu)	238.38 ± 0.24	**233.76** ± 0.09	**153.43** ± 0.15	17.5 ± 0.04	**16.3** ± 0.01	**14.3** ± 0.01
Glycine (Gly)	52.75 ± 0.08	**55.31** ± 0.10	**42.40** ± 0.08	3.9 ± 0.01	3.9 ± 0.01	3.9 ± 0.01
Histidine (His)	21.13 ± 0.03	21.95 ± 0.06	**15.79** ± 0.04	1.5 ± 0.00	1.5 ± 0.00	1.5 ± 0.00
Isoleucine (Ile)	39.86 ± 0.07	**41.29** ± 0.06	**32.56** ± 0.02	2.9 ± 0.01	2.9 ± 0.00	**3.0** ± 0.00
Leucine (Leu)	63.08 ± 0.10	**68.34** ± 0.06	**53.01** ± 0.03	4.6 ± 0.01	**4.8** ± 0.01	**4.9** ± 0.00
Lysine (Lys)	23.40 ± 0.04	**25.35** ± 0.02	**19.66** ± 0.01	1.7 ± 0.00	**1.8** ± 0.00	**1.8** ± 0.00
Methionine (Met)	192.00 ± 0.40	**228.00** ± 0.67	**199.26** ± 0.43	14.1 ± 0.04	**15.9** ± 0.05	**18.5** ± 0.04
Phenylalanine (Phe)	45.60 ± 0.09	46.01 ± 0.15	**32.87** ± 0.08	3.3 ± 0.01	**3.2** ± 0.01	**3.1** ± 0.01
Proline (Pro)	40.28 ± 0.15	**47.85** ± 0.03	**30.02** ± 0.00	3.0 ± 0.01	**3.3** ± 0.00	**2.8** ± 0.00
Serine (Ser)	63.98 ± 0.12	64.75 ± 0.20	**48.68** ± 0.15	4.7 ± 0.01	**4.5** ± 0.01	**4.5** ± 0.01
Threonine (Thr)	61.98 ± 0.26	**71.14** ± 0.05	**46.89** ± 0.06	4.5 ± 0.02	**5.0** ± 0.00	**4.4** ± 0.01
Tryptophan (Trp)	104.36 ± 0.90	**61.91** ± 0.75	**69.03** ± 0.57	7.6 ± 0.07	**4.3** ± 0.05	**6.4** ± 0.05
Tyrosine (Tyr)	48.70 ± 0.07	48.46 ± 0.15	**35.74** ± 0.11	3.6 ± 0.01	**3.4** ± 0.01	**3.3** ± 0.01
Valine (Val)	51.26 ± 0.09	**52.94** ± 0.07	**42.44** ± 0.01	3.8 ± 0.01	**3.7** ± 0.01	**3.9** ± 0.00
Total AA	1364.91 ± 2.46	**1431.74** ± 0.98	**1075.43** ± 0.41	100 ± 0.25	100 ± 0.10	100 ± 0.05

**Table 3 animals-13-02812-t003:** Estimated amino acid (AA) absorption during the gastrointestinal digestion, expressed as absolute values (mg/reactor, corresponding to mg/3.2 g product) and relative proportion (% compared to total protein absorbed and % compared to individual AA content in raw test product), of different products (drum-dried torula yeast broth produced on dextrose (DDI), drum-dried torula yeast broth produced on wood-derived hydrolysates (WDI), and spray-dried torula yeast broth produced on wood-derived hydrolysates (WSI)). Results are shown as averages ± stdev (*n* = 3). Statistical analysis was performed to compare the different test products using a two-way ANOVA with Tukey’s multiple comparisons post hoc test. Statistical differences between DDI and WDI, and DDI and WSI, are indicated in bold, while significant differences between WDI and WSI are underlined (*p* < 0.05).

	Absorbed AA (mg/Reactor; 3.2 g Product)			% AA Absorbed (to Total Protein Absorbed)			% AA Absorbed (to Individual AA in Raw Product)		
	DDI	WDI	WSI	DDI	WDI	WSI	DDI	WDI	WSI
Alanine (Ala)	55.3 ± 1.1	69.9 ± 5.5	**26.5** ± 2.0	7.0 ± 0.3	9.3 ± 0.8	6.0 ± 0.8	58.7 ± 1.2	60.0 ± 4.7	**38.1** ± 2.8
Arginine (Arg)	36.2 ± 1.3	34.2 ± 0.3	**14.9** ± 0.4	4.6 ± 0.2	4.6 ± 0.2	3.4 ± 0.4	61.6 ± 2.2	56.0 ± 0.5	**39.0** ± 1.1
Aspartic acid (Asp)	53.9 ± 1.0	47.7 ± 11.6	**27.3** ± 2.4	6.8 ± 0.2	6.4 ± 1.6	6.1 ± 0.9	41.9 ± 0.8	35.0 ± 8.5	**25.4** ± 2.2
Cysteine (Cys)	50.4 ± 6.9	39.6 ± 4.5	**29.9** ± 5.0	6.4 ± 0.9	5.3 ± 0.6	6.7 ± 1.4	70.5 ± 9.7	78.0 ± 8.9	77.7 ± 13.1
Glutamic acid (Glu)	135.5 ± 2.3	**114.2** ± 13.2	**50.6** ± 4.1	17.2 ± 0.6	15.2 ± 1.9	**11.4** ± 1.5	56.8 ± 1.0	48.8 ± 5.6	**33.0** ± 2.7
Glycine (Gly)	25.2 ± 1.3	21.3 ± 0.8	12.9 ± 0.6	3.2 ± 0.2	2.8 ± 0.2	2.9 ± 0.3	47.7 ± 2.4	**38.5** ± 1.5	**30.3 **± 1.5
Histidine (His)	9.9 ± 0.7	8.2 ± 0.8	3.6 ± 0.1	1.3 ± 0.1	1.1 ± 0.1	0.8 ± 0.1	47.1 ± 3.4	**37.5** ± 3.8	**23.0** ± 0.9
Isoleucine (Ile)	13.1 ± 0.4	9.4 ± 2.5	4.0 ± 1.1	1.7 ± 0.1	1.3 ± 0.3	0.9 ± 0.3	33.1 ± 1.1	**22.7** ± 6.0	**12.3** ± 3.2
Leucine (Leu)	29.1 ± 1.1	27.7 ± 3.3	15.7 ± 1.5	3.7 ± 0.2	3.7 ± 0.5	3.5 ± 0.5	46.2 ± 1.7	40.5 ± 4.8	**29.6** ± 2.9
Lysine (Lys)	10.3 ± 0.4	9.9 ± 2.0	5.7 ± 1.0	1.3 ± 0.1	1.3 ± 0.3	1.3 ± 0.3	44.2 ± 1.5	39.1 ± 7.7	**28.9 **± 4.8
Methionine (Met)	148.7 ± 4.1	**176.7** ± 7.7	**155.4** ± 2.1	18.9 ± 0.8	**23.6** ± 1.4	**34.9** ± 3.8	77.5 ± 2.1	77.5 ± 3.4	78.0 ± 1.0
Phenylalanine (Phe)	24.9 ± 1.8	21.2 ± 1.0	9.0 ± 0.3	3.2 ± 0.2	2.8 ± 0.2	2.0 ± 0.2	54.6 ± 3.9	**46.0** ± 2.1	**27.5** ± 0.9
Proline (Pro)	17.2 ± 0.7	19.9 ± 2.1	5.7 ± 0.4	2.2 ± 0.1	2.7 ± 0.3	1.3 ± 0.2	42.8 ± 1.8	41.7 ± 4.5	**19.1** ± 1.3
Serine (Ser)	32.7 ± 1.8	25.3 ± 3.1	**13.5** ± 1.4	4.1 ± 0.3	3.4 ± 0.4	3.0 ± 0.5	51.1 ± 2.8	**39.1** ± 4.9	**27.6** ± 2.9
Threonine (Thr)	24.2 ± 1.2	24.2 ± 0.2	**3.5** ± 3.6	3.1 ± 0.2	3.2 ± 0.1	0.8 ± 0.8	39.0 ± 2.0	34.0 ± 0.3	**7.4** ± 7.7
Tryptophan (Trp)	104.4 ± 0.9	**61.9** ± 0.7	**25.8** ± 0.6	13.2 ± 0.4	**8.3** ± 0.3	**5.8** ± 0.6	100.0 ± 0.9	100.0 ± 1.2	**37.4** ± 0.8
Tyrosine (Tyr)	28.8 ± 1.7	25.7 ± 1.5	13.7 ± 0.1	3.7 ± 0.2	3.4 ± 0.2	3.1 ± 0.3	59.2 ± 3.5	53.1 ± 3.2	**38.4** ± 0.4
Valine (Val)	17.8 ± 0.7	**12.4** ± 3.2	**5.9** ± 1.7	2.3 ± 0.1	1.7 ± 0.4	1.3 ± 0.4	34.8 ± 1.3	**23.5** ± 6.0	**14.0** ± 4.0
Total AA	787.9 ± 24.2	749.3 ± 28.3	445.2 ± 48.7	100.0 ± 4.3	100.0 ± 5.3	100.0 ± 15.5	57.7 ± 1.8	52.3 ± 2.0	**41.4** ± 4.5

## Data Availability

The data that support the findings of this study are displayed in the manuscript. Any further information about data is available from the corresponding author, M.M, upon reasonable request.

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
