# Peer review of "Amino Acid Digestibility of Different Formulations of Torula Yeast in an In Vitro Porcine Gastrointestinal Digestion Model and Their Protective Effects on Barrier Function and Inflammation in a Caco-2/THP1Co-Culture Model"

_animals, 2023, doi:10.3390/ani13182812_

Round 1

Reviewer 1 Report

This is an interesting topic on alternative protein sources for animal nutrition. In general, the authors need to justify the use of the Caco2 cells instead of Swine intestinal cell lines for the study and better link the study to swine. Below are my comments and suggestions:

Title: the title in not explicit enough: what about the intestinal cell model used?

In this case the study used Caco2 cells. In addition, if the study is about a specific species, the title has to show it.

Abstract: One of the objectives of this study was to evaluate the amino acid digestibility and absorption, there are no result regarding these two aspects in the abstract. The authors need to present some summary of the results on digestibility.  Line 29-30: which products showed 41-58% bio-accessibility?

Introduction:

Line 57-58: this topic sentence does not relate to the content of the paragraph and should be revised.

Line 68: I don’t understand what the authors mean by “control of epithelial barrier”. Please be specific.

Line 85-86: being the AA …. This sentence needs to be revised for clarity.

Line 105: this is the first time the c. jadinii is mentioned. Please give complete name first.

Line 113-116: I think this is far reaching by relating the study to intestinal health in pigs. The study used Caco2cell from humans and the link to pigs is unclear.

Materials and methods:

It is unclear why the authors chose to use Caco-2 cells, which are cell line from human colon; however, in the introduction the authors their objectives to swine. The authors need to clarify why they didn’t use porcine intestinal epithelial cell line, which are even more specific to the small intestine for their study.  

Results:

Line 277-309: did the authors perform statistical analysis on the parameters described in this paragraphs? There is no mention of p values, and the values mentioned are already in the tables. For examples, see in line 278, 42.7g/100g, 44.7g/100g, and 33.6g/100g. The reader needs to see what the differences are. Please, revise the result of this section lines 277-309 accordingly.

Tables:

Table 2, as is table 2 doesn’t tell anything. The table should show the mean separations. I suggest combining table 2 and tables s2 and S3 into one or two tables that show the mean separation, so that we can clearly see the differences among the treatments (DDI, WDI, and WSI) for each amino acid.  in addition, include all the tables directly in the document.

Table 3: did the authors perform statistical analysis on the parameters presented in table 3? if statistical analysis was performed, then the table 3 need to show the p values and mean separations similar to table 2.

In general, the tables have to stand alone. Please add footnote defining treatments and abbreviations and stating significance.

 Are the data presented in figure 1 descriptive? If that is the case, please state it clearly. However, if the authors have performed statistical analysis, then they need to add p values and significance. In this case I suggest using clustered column instead of stacked ones.

Line 354: define GIT.

Discussion:

All the discussion is on pigs intestinal health, and I don’t see the link between the experimental protocol, especially the use of Caco2 cell and pigs intestinal health. The authors need to clearly why they use Caco2 cell, how the results can be translated to pigs, and revise the discussion accordingly.   

Line 435: DDI has lower fat…. Did the authors determine fat content or is this a data from another study?

Line 498: the study was not conducted in pigs (in vivo) or in vitro using swine intestinal cell line (for example, swine intestinal IPEC-j2 cells), therefore I think the last sentence (provided new insights into….of pigs fed torula yeast) of the discussion is too strong and overreaching.  

Reviewer 2 Report

This manuscript is to investigate the AA digestibility of different type of torula yeast with in vitro gastrointestinal simulation, and the protective effects of digesta on the integrity and immune response of intestinal cells. The results showed that there was difference for digestibility among three torula yeast, and torula yeast digesta protected the intestinal barrier from inflammatory damage.

1. The title of the manuscript does not include all the content of the study, there is a major portion for the AA digestibility of different type of torula yeast which is not included in the title.

2. The authors used in vitro simulation to detect the AA digestibility of torula yeast, however, it did not provide the basis for the composition and concentration of simulated salivary fluid, gastric fluid, and intestinal fluid. To mimic the digestion of torula yeast in pig gastrointestinal tract, which stage of pigs for mimic? weaning pigs, growing pigs or finishing pigs?

3. The concentration of Met and Trp are too high in the torula yeast, the method for detection AA need to check. However, the digestibility of AA is much lower to compare with in vivo study (Stein, 2020).

4. Although it is found that digesta is beneficial for the integrity and immune response of epithelia cells, it is not known which components have these effect.

5. The author conducted this study to provide the information for torula yeast on pig production, however, the in vitro digestion is not mention it is mimic the GI tract of pigs, the cell line used is from human not from pigs, so it is hard to say this study will give important values for pig.

Round 2

Reviewer 1 Report

Figure 1 represents the results of two different analysis. I think the authors should separate in two different graphs absorbed (fig 1a) and non-absorbed (fig1b) for more clarity.

Author Response

Dear Reviewer 

Many thanks for this additional comment. However, the data in figure 1 do belong to the same experiment and type of analysis. Here, it is shown the fraction of amino acids that are absorbed at the end of the small intestine vs the non-absorbed fraction. The two set of results are complementary and hence we do believe that a single figure (in place of fig. 1a and 1b) it is of easier interpretation for the reader.

Hope you will agree with us.

Best regards

Massimo Marzorati